# LTD-Bench: Evaluating Large Language Models by Letting Them Draw

**Liuhao Lin**[*†]
Key Laboratory of Multimedia
Trusted Perception and Efficient
Computing, Ministry of Education
of China, Xiamen University
Xiamen, China

**Ke Li**[†‡]
Tencent Youtu Lab
Shanghai, China

**Zihan Xu**
Tencent Youtu Lab
Shanghai, China

**Yuchen Shi**
Tencent Youtu Lab
Shanghai, China

**Yulei Qin**
Tencent Youtu Lab
Shanghai, China

**Yan Zhang**[§]
Key Laboratory of Multimedia
Trusted Perception and Efficient
Computing, Ministry of Education
of China, Xiamen University
Xiamen, China

**Xing Sun**
Tencent Youtu Lab
Shanghai, China

**Rongrong Ji**
Key Laboratory of Multimedia
Trusted Perception and Efficient
Computing, Ministry of Education
of China, Xiamen University
Xiamen, China

## Abstract

Current evaluation paradigms for large language models (LLMs) represent a critical blind spot in AI research—relying on opaque numerical metrics that conceal fundamental limitations in spatial reasoning while providing no intuitive understanding of model capabilities. This deficiency creates a dangerous disconnect between reported performance and practical abilities, particularly for applications requiring physical world understanding. We introduce LTD-Bench, a breakthrough benchmark that transforms LLM evaluation from abstract scores to directly observable visual outputs by requiring models to generate drawings through dot matrices or executable code. This approach makes spatial reasoning limitations immediately apparent even to non-experts, bridging the fundamental gap between statistical performance and intuitive assessment. LTD-Bench implements a comprehensive methodology with complementary generation tasks (testing spatial imagination) and recognition tasks (assessing spatial perception) across three progressively challenging difficulty levels, methodically evaluating both directions of the critical language-spatial mapping. Our extensive experiments with state-of-the-art models expose an alarming capability gap: even LLMs achieving impressive results on traditional benchmarks demonstrate profound deficiencies in establishing bidirectional mappings between language and spatial concepts—a fundamental limitation that undermines their potential as genuine world models. Furthermore, LTD-Bench's visual outputs enable powerful diagnostic analysis, offering a poten-

---

[*]This work was done during an internship at Tencent Youtu Lab.

[†]These authors have contributed equally.

[‡]Project lead.

[§]Corresponding authors.

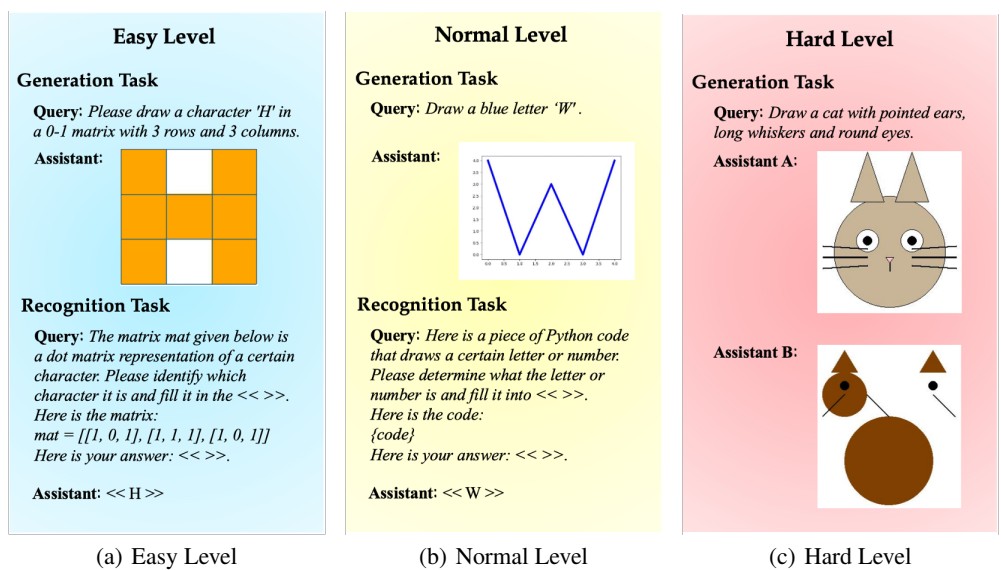

| Easy Level | Normal Level | Hard Level |

**Figure 1:** The data examples of three levels in LTD-Bench. The model outputs in the generation tasks have all been rendered into images.

tial approach to investigate model similarity. Our dataset and codes are available at
`https://github.com/walktaster/LTD-Bench`

# 1 Introduction

Large language models (LLMs) have demonstrated remarkable progress in recent years [26][25][3], achieving impressive results on numerous benchmarks spanning language understanding [14][17], mathematical reasoning [8][15][7][6], code generation [4][2] and instruction following [33][16][20]. However, these apparent successes mask a critical blind spot: current evaluation paradigms, which rely heavily on aggregate scores and opaque metrics, offer limited insight into models' true capabilities—particularly their understanding of the physical world. This disconnect is especially concerning as LLMs are increasingly deployed in domains such as robotics, autonomous systems, and design tools [30][24][28][21], where spatial reasoning is essential.

What makes this problem particularly pernicious is the abstract nature of traditional evaluation. When a model scores 85% on a benchmark, what specific capabilities and limitations does this number reveal? How can researchers, developers, and end-users gain an intuitive understanding of what the model can and cannot do in spatial domains? These questions remain largely unanswered by current methodologies.

To address this gap, we introduce **LTD-Bench** (Let Them Draw Benchmark), a novel evaluation framework that shifts LLM assessment from abstract numerical scores to directly observable visual outputs. Figure 1 shows the data examples of LTD-Bench. Unlike conventional benchmarks, LTD-Bench requires models to generate visual artifacts—either as dot matrices or executable code—based on textual instructions, making their spatial reasoning abilities immediately apparent even to non-experts. This approach bridges the disconnect between statistical metrics and intuitive understanding of model capabilities.

LTD-Bench comprises two complementary evaluation paths: generation tasks, which assess spatial imagination by requiring models to translate textual descriptions into visual representations, and recognition tasks, which evaluate spatial perception by asking models to interpret visual patterns. These tasks span three progressively challenging levels, from basic character representation to complex real-world object visualization, enabling fine-grained analysis of spatial reasoning abilities.

Our experiments with state-of-the-art LLMs reveal a significant capability gap overlooked by existing benchmarks. Even models that perform well on traditional reasoning tasks exhibit profound deficien-

cies in mapping between language and spatial concepts, undermining their potential as genuine world models. Furthermore, LTD-Bench's visual outputs facilitate diagnostic analyses not possible with traditional benchmarks, such as comparing stylistic characteristics of generated images to investigate model similarities.

By making model limitations visible rather than obscured behind abstract metrics, LTD-Bench represents a paradigm shift in the evaluation and understanding of large language models. Our framework lays the foundation for developing AI systems with more robust spatial reasoning—a critical requirement for applications that must interact with and reason about the physical world.

Our contributions are summarized as follows:

- We introduce LTD-Bench, the first benchmark that transforms LLM evaluation from opaque metrics to visually interpretable outputs. By requiring models to generate visual artifacts through drawing, we enable direct human assessment of spatial reasoning capabilities, addressing a fundamental gap between statistical performance and intuitive understanding of model limitations.
- We design a structured evaluation methodology with complementary generation and recognition tasks across three difficulty levels, providing a comprehensive assessment of both how LLMs translate language into spatial arrangements (imagination) and interpret spatial patterns into language (perception).
- Our experimental results quantify a significant capability gap in current LLMs, showing that even models with strong reasoning abilities struggle to establish the bidirectional mapping between language and spatial concepts - a critical finding that identifies a priority direction for improvement in the next generation of AI systems.
- We demonstrate how visual output comparison provides a powerful diagnostic tool for model development, revealing stylistic similarities among various models and offering insights into the model similarity that is not well captured by traditional evaluation metrics.

## 2 Related Work and Discussion

**Existing Benchmarks.** Current LLM evaluation frameworks primarily emphasize symbolic and procedural competencies. Comprehensive benchmarks such as MMLU [14] and TruthfulQA [17] assess cross-domain knowledge retention and factual accuracy, while mathematical reasoning datasets like GSM8K [8] and MATH [15] focus on multi-step problem-solving in abstract domains. Code generation benchmarks (e.g., HumanEval [4], MBPP [2]) further evaluate the translation of natural language into executable algorithms. While these benchmarks effectively quantify core symbolic manipulation skills, they are confined to a text-to-symbol paradigm and lack intuitive, visual, and directly interpretable assessments of model capabilities. Consequently, they do not reveal whether LLMs can establish robust, bidirectional mappings between linguistic symbols and spatial entities.

**Spatial Perception and Imagination.** The lack of spatial evaluation in LLMs is partly rooted in the assumption that visual perception is essential for spatial reasoning. However, neurocognitive studies of congenitally blind individuals demonstrate that robust spatial cognition can arise through nonvisual modalities, such as linguistic descriptions and haptic feedback. For instance, Striem-Amit et al. [22] provide evidence for a neural dissociation between abstract semantic knowledge and sensory attributes, while Cooney et al. [9] show that spatial reasoning relies on innate neural mechanisms rather than visual experience. These findings challenge the necessity of visual input for spatial reasoning and establish a biological precedent for text-based spatial cognition. This suggests that text-based LLMs, even without visual input, should be capable of developing spatial understanding. Moreover, Transformer architectures, which underpin modern LLMs, excel at modeling relationships between abstract tokens, theoretically enabling the inference of geometric and topological patterns from textual descriptions. Recent work supports this potential: LLMs have demonstrated the ability to generate code for rendering simple shapes [1][13], indicating latent spatial reasoning capabilities. Nevertheless, no existing benchmark systematically evaluates these abilities, despite their importance for meaningful interaction with the physical world.

**Intuitive Visual Evaluation.** LTD-Bench addresses this gap by providing an intuitive and visual assessment of LLMs' spatial perception and imagination. In generation tasks, models are required

to produce either renderable dot matrices or Python code for image drawing, both of which can be visualized as images. Prior research has shown that prompting LLMs for evaluation is effective not only in NLP tasks [32][5][19][10] but also in multimodal domains [31][29]. Accordingly, LTD-Bench leverages GPT-4.1 to assess the quality of images generated by LLMs in certain open-ended generative tasks, enabling a more direct and interpretable evaluation of spatial reasoning abilities.

# 3 LTD-Bench

The gap between reported LLM performance and actual spatial reasoning capabilities represents a critical blind spot in AI evaluation. LTD-Bench addresses this gap through a novel approach that transforms abstract metrics into directly observable visual evidence, enabling intuitive assessment of how well models can establish bidirectional mappings between language and spatial concepts.

## 3.1 Design Principles and Problem Addressing

LTD-Bench addresses three fundamental problems in current LLM evaluation approaches through carefully designed principles:

**Problem 1: Invisibility of Spatial Reasoning Limitations.** Current benchmarks provide numerical scores that obscure whether models can actually establish bidirectional mappings between language and spatial concepts.

**Solution: Visual Interpretability.** LTD-Bench's core innovation is its transformation of abstract model capabilities into concrete visual artifacts. All outputs from generation tasks are rendered into images, enabling direct inspection by both humans and automated systems. This approach makes model limitations immediately apparent to anyone - regardless of technical background - revealing capabilities that remain hidden in traditional benchmarks.

**Problem 2: Incomplete Assessment of Spatial Cognition.** Existing evaluations rarely assess both directions of the critical language-spatial mapping.

**Solution: Dual-Path Evaluation.** LTD-Bench systematically evaluates both aspects of spatial cognition through complementary pathways:

- **Generation Tasks (Spatial Imagination):** Models translate textual descriptions into visual representations (dot matrices or drawing code), testing their ability to convert linguistic concepts into spatial arrangements.

- **Recognition Tasks (Spatial Perception):** Models interpret visual patterns from given representations, testing their ability to understand spatial configurations through language.

**Problem 3: Inability to Pinpoint Capability Thresholds.** Traditional benchmarks often fail to identify precisely where models begin to struggle with increasingly complex spatial reasoning.

**Solution: Progressive Complexity.** LTD-Bench implements a hierarchical structure with three difficulty levels:

1. **Easy Level:** Basic character representation using discrete dot matrices in finite grid space, establishing baseline spatial capabilities.

2. **Normal Level:** Character drawing using continuous curves in infinite coordinate space, requiring more sophisticated spatial reasoning.

3. **Hard Level:** Complex real-world object representation, requiring advanced spatial conceptualization and compositional understanding.

Through these design principles, LTD-Bench not only evaluates model performance but fundamentally transforms how we understand and interpret model capabilities, making limitations visible that were previously hidden behind abstract metrics.

Table 1: The structure of LTD-Bench

| Level | Task | | Total |
| | Generation | Recognition | |
|---|---|---|---|
| Easy | 50 | 36 | 86 |
| Normal | 36 | 36 | 72 |
| Hard | 25 | - | 25 |
| Total | 111 | 72 | 183 |

## 3.2 Benchmark Structure and Task Design

Based on the design principles outlined in Section 3.1, LTD-Bench comprises a comprehensive evaluation framework with 183 distinct data distributed across three difficulty levels, as summarized in Table 1. Each level presents unique challenges designed to assess specific aspects of spatial reasoning.

### 3.2.1 Easy Level: Discrete Grid-Based Spatial Understanding

The Easy level is designed to assess the fundamental spatial abilities of LLMs within a two-dimensional finite grid space represented in the form of a dot matrix. In accordance with the dual-path evaluation principle, this level consists of both generation and recognition tasks:

**Generation Task:** Given a textual instruction (e.g. "Please draw a character 'H' in a 0-1 matrix with 3 rows and 3 columns."), the model is required to output a dot matrix, where '1' denotes a filled cell and '0' denotes a blank cell, representing the specified character. The output dot matrix can be rendered into a grid-like image for intuitive and visual display.

This task evaluates whether models can:

- Conceptualize the spatial arrangement of simple characters
- Translate this conceptualization into a precise grid-based representation
- Maintain correct proportions and spatial relationships within constraints

**Recognition Task:** In the complementary recognition pathway, models are presented with a dot matrix and must identify which character it represents. This evaluates the reverse mapping—from spatial arrangement to symbolic representation—testing models' ability to interpret visual patterns expressed as matrices.

As shown in Figure 1(a), the outputs of generation tasks at this level can be directly verified through visual inspection, while recognition tasks have unambiguous ground-truth answers. This level establishes whether models possess even the most basic capacity for bidirectional mapping between language and discrete spatial representations.

### 3.2.2 Normal Level: Curve Composition in Infinite 2D Coordinate Space

The Normal level increases complexity by transitioning from discrete grid spaces to continuous coordinate systems, requiring models to reason about characters as compositions of mathematical curves in an unbounded space.

**Generation Task:** The model is tasked with generating Python code that draws specified characters (such as letters or digits) using only combinations of curves. Prompts are designed to enforce the constraint that only curve-based drawing methods are allowed, explicitly prohibiting the use of direct text rendering functions (e.g., TextPrint). The generated Python code can be executed directly to produce images, enabling an intuitive and visual evaluation of the correctness of the model's drawings.

This level evaluates whether models can:

- Translate character concepts into continuous rather than discrete representations
- Generate executable code that correctly implements spatial understanding

- Create visual outputs that maintain character recognizability despite implementation constraints

**Recognition Task:** For recognition, models are presented with Python code that draws a character through curve combinations and must identify which character the code would render. This requires parsing code, understanding how mathematical functions translate to visual shapes, and recognizing the resulting pattern.

As illustrated in Figure 1(b), this level significantly increases the complexity of the bidirectional mapping between language and spatial concepts, requiring models to operate in continuous rather than discrete space and to translate between linguistic, mathematical and visual representations.

### 3.2.3 Hard Level: Real-world Object Drawing in Infinite 2D Space

The Hard level represents the most advanced spatial reasoning challenge, requiring models to conceptualize and render complex real-world objects as compositions of multiple curves.

**Generation Task:** Models receive open-ended instructions to draw real-world objects with specific attributes, such as "Draw a cat with pointed ears, long whiskers and round eyes." This requires not only understanding what these objects look like but also decomposing them into geometric primitives and implementing them through code.

This level evaluates whether models can:

- Conceptualize complex multi-part objects from linguistic descriptions
- Translate abstract object features into concrete spatial relationships
- Generate code that produces a coherent and recognizable visual representation

**Evaluation Approach.**  Due to the open-ended and subjective nature of these tasks, evaluation at this level employs GPT-4.1 as an automated assessor following established practices in LLM-based evaluation [29][32][5][19][10]. The evaluation protocol assigns scores between 0.0 and 1.0 based on predefined criteria that consider both adherence to specified features and overall visual coherence.

Additionally, as shown in Figure 1(c), the stylistic diversity of outputs at this level enables comparative analysis between different model architectures. By examining stylistic similarities in generated images, researchers can identify shared tendencies across model families, providing insights into the similarity among various LLMs that is not captured by traditional evaluation metrics.

Together, these three levels form a comprehensive framework for evaluating spatial perception and imagination capabilities in LLMs, making visible the specific strengths and limitations that remain hidden in traditional text-based benchmarks.

## 4 Experiments

### 4.1 Experiment Settings

**Models.**  Since our approach places certain demands on both the reasoning and coding abilities of LLMs, and its tasks are relatively challenging for smaller LLMs, we primarily selected some of the most advanced models for evaluation, including DeepSeek-R1 [12], DeepSeek-V3 [18], GPT-4o, GPT-4.1-mini, QwQ32B [23], Qwen2.5-72B-Instruct [27] and Llama3.3-70B-Instruct [11].

**Evaluation Methods.**  The evaluation of our benchmark is tailored to the nature of each task type and difficulty level.

For generation tasks, which lack fixed ground-truth answers, we employ both human evaluation and GPT-4.1-based automated evaluation at the Easy and Normal levels. An analytical comparison between human and GPT-4.1 evaluations is provided in Appendix A.3. For Hard-level generation tasks, the open-ended nature of the outputs introduces considerable subjectivity into human assessment, as personal aesthetic preferences and other factors may influence scoring. Therefore, we rely exclusively on GPT-4.1 for evaluation at this level, utilizing a detailed system prompt with explicit scoring criteria to guide GPT-4.1 in assigning a score between 0.0 and 1.0 to each generated image.

Table 2: The model output examples on the generation tasks across different difficulty levels in LTD-Bench. All outputs are rendered as images to facilitate an intuitive visual assessment of the model's capabilities.

| | Easy Generation *Please draw a character 'K' in a 0-1 matrix with 5 rows and 4 columns* | Normal Generation *Draw a green letter Q* | Hard Generation *Draw a cat with pointed ears, long whiskers and round eyes* |
|---|---|---|---|
| Deepseek-r1 | | | |
| Deepseek-v3 | | | |
| GPT-4.1-mini | | | |
| GPT-4o | | | |
| QwQ-32B | | | |
| Qwen2.5-72B-Instruct | | | |
| Llama3.3-70B-Instruct | | | |

If a model's output code fails to execute and does not produce a valid image, a score of 0 is assigned for that instance.

For recognition tasks, which have well-defined ground-truth answers, we directly compare model outputs to the correct answers to compute accuracy.

More implementation details are provided in Appendix A.1, and the prompt templates used in LTD-Bench are included in Appendix B.

## 4.2 Result Analyses

Since the intuitive visual evaluation of LLM capabilities is a central contribution of our work, we first present representative model outputs for generation tasks across different difficulty levels in Table 2. These examples clearly illustrate performance disparities among models. For instance, advanced models such as Deepseek-r1 and GPT-4.1-mini significantly outperform smaller models like Qwen2.5-72B-Instruct and Llama3.3-70B-Instruct. Overall performance metrics are summarized in Table 3, from which we draw the following conclusions:

**LLMs generally exhibit poor spatial perception and imagination.** As shown in Table 3, only Deepseek-r1 achieves an average accuracy above 70%, with GPT-4.1-mini exceeding 60%. In contrast, Qwen2.5-72B-Instruct and Llama3.3-70B-Instruct achieve only around 30%. Notably, human experts can solve Easy and Normal tasks with near-perfect accuracy, even in text-only settings, whereas LLMs fall far short. These results indicate that current LLMs lack robust spatial reasoning and fail to

Table 3: LTD-Bench evaluation results. **Bold** indicates the best performance in that dimension, while underline indicates the second-best performance. For generation tasks at Easy and Normal level, the data in blue is the results evaluated by human and data in orange is the results evaluated by GPT-4.1. Numbers are presented in % with a full score of 100%.

| Model | Easy | | Normal | | Hard | Average |
|-------|------------|-------------|------------|-------------|------------|---------|
| | Generation | Recognition | Generation | Recognition | Generation | |
| Deepseek-r1 | 82.00 (80.00 / 84.00) | **69.44** | 65.28 (55.56 / 75.00) | **77.78** | 63.20 | **71.54** |
| Deepseek-v3 | 72.00 (66.00 / 78.00) | 36.11 | 54.17 (47.22 / 61.11) | 63.89 | 66.40 | 58.51 |
| GPT-4.1-mini | **85.00 (82.00 / 88.00)** | 38.89 | **70.83 (66.67 / 75.00)** | 55.56 | **71.60** | 64.38 |
| GPT-4o | 81.00 (76.00 / 86.00) | 41.67 | 45.83 (36.11 / 55.56) | 44.44 | 48.00 | 52.19 |
| QwQ-32B | 65.00 (58.00 / 72.00) | 36.11 | 38.89 (33.33 / 44.44) | 58.33 | 42.00 | 48.07 |
| Qwen2.5-72B-Instruct | 56.00 (42.00 / 70.00) | 13.89 | 18.06 (13.89 / 22.22) | 25.00 | 40.80 | 30.75 |
| Llama3.3-70B-Instruct | 46.00 (32.00 / 60.00) | 11.11 | 23.61 (16.67 / 30.56) | 19.44 | 35.20 | 27.07 |

Table 4: Model performance on generation and recognition tasks. **Bold** indicates the best performance in that dimension, while underline indicates the second-best performance.

| Model | Generation | Recognition | Average |
|-------|------------|-------------|---------|
| Deepseek-r1 | 72.88 | **73.61** | **73.24** |
| Deepseek-v3 | 64.71 | 50.00 | 57.36 |
| GPT-4.1-mini | **77.46** | 47.22 | 62.34 |
| GPT-4o | 62.07 | 43.06 | 52.56 |
| QwQ-32B | 51.33 | 47.22 | 49.28 |
| Qwen2.5-72B-Instruct | 39.17 | 19.44 | 29.31 |
| Llama3.3-70B-Instruct | 36.62 | 15.28 | 25.95 |

establish reliable bidirectional mappings between linguistic symbols and spatial entities—an essential capability for genuine world comprehension. Further analysis is provided in Appendix C.

**Deep reasoning improves recognition but not generation tasks.** Table 4 shows that Deepseek-r1, equipped with deep reasoning, outperforms GPT-4.1-mini by over 25% in recognition accuracy, but lags behind in generation tasks. Similarly, within the Deepseek family, Deepseek-r1 consistently surpasses Deepseek-v3, yet the relative improvement is much less pronounced for generation than for recognition. QwQ-32B, another model with deep reasoning, matches GPT-4.1-mini on recognition but underperforms on generation. We hypothesize that recognition tasks benefit from enhanced spatial perception via reasoning, while generation tasks, which rely more on spatial imagination, are less amenable to such improvements.

Table 5 further supports this: Llama3.3-70B distilled with Deepseek-r1 data shows an 18.05% accuracy gain on recognition tasks, but even a 2.91% decline on generation tasks. This suggests that deep reasoning may lead to overthinking in tasks where LLMs' inherent abilities are insufficient, potentially degrading performance.

**Multimodal LLMs do not show clear advantages on text-based spatial tasks.** Contrary to human intuition—where visual experience enhances spatial abilities—multimodal models like GPT-4.1-mini and GPT-4o do not consistently outperform text-only models such as Deepseek-r1 and Deepseek-v3 on LTD-Bench (Tables 3 and 4). While GPT-4.1-mini excels in generation tasks, its overall performance is still lower than Deepseek-r1, and GPT-4o underperforms compared to Deepseek-v3. Although these results may be influenced by differences in language modeling capabilities, they challenge expectations based on human cognition and highlight the need for further research on aligning visual and textual features in multimodal learning.

Table 5: Comparison of the performance of Llama3.3-70B-Instruct and Deepseek -r1-distill-Llama3.3-70B on generation and recognition tasks.

| Model | Generation | Recognition | Average |
|---|---|---|---|
| Llama3.3-70B-Instruct | 36.62 | 15.28 | 25.95 |
| Deepseek-r1-distill-Llama3.3-70B | 33.71 | 33.33 | 33.52 |
| Δ | 2.91↓ | 18.05↑ | 7.57↑ |

Table 6: Style similarity comparison on the generation task at Hard level. (1) is Qwen2.5-72B-Instruct, (2) is Qwen2.5-32B-Instruct and (3) is GPT-4.1-mini. The total sample size is 22, excluding the images that failed to be generated.

| | (1) and (2) more similar | (1) and (3) more similar | (2) and (3) more similar | All three are different |
|---|---|---|---|---|
| Rate | 12/22 | 1/22 | 2/22 | 7/22 |

## 4.3 Model Similarity

For Hard-level generation tasks, different models tend to produce images with distinct stylistic characteristics. Analyzing the stylistic similarity among these images provides a promising avenue for investigating model similarity. As shown in Table 6, we conduct a style similarity comparison among three models: Qwen2.5-72B-Instruct, Qwen2.5-32B-Instruct, and GPT-4.1-mini. The results indicate that the highest proportion of stylistically similar images occurs between the two models from the Qwen2.5 series, accounting for over 50% of the total valid samples. In contrast, only three samples were found to be more similar to GPT-4.1-mini. Table 2 further presents representative examples, offering a more intuitive illustration of the stylistic similarities among images generated by the three models. It is evident from these examples that the Qwen2.5 series models exhibit greater stylistic resemblance to each other. This suggests a positive correlation between model similarity and the stylistic similarity of the open-ended images they generate. Through this exploratory study, we identify a potential new approach for assessing model similarity, which may serve as a valuable reference for future research in this area.

## 5 Limitations and Future Work

While LTD-Bench offers an intuitive and visual framework for evaluating LLM capabilities, several limitations remain. First, the current benchmark features a relatively small dataset and assesses a narrow range of abilities, focusing solely on spatial perception and imagination. This limits both the comprehensiveness and generalizability of our findings. Future work will expand the dataset and incorporate a wider array of tasks to enable more thorough evaluation of LLMs.

Table 7: The specific image examples generated by Qwen2.5-72B-Instruct, Qwen2.5-32B-Instruct and GPT-4.1-mini.

| | Qwen2.5-72B-Instruct | Qwen2.5-32B-Instruct | GPT-4.1-mini |
|---|---|---|---|
| *Flower* | | | |
| *House* | | | |
| *Rabbit* | | | |

Second, our analysis of model similarity is preliminary, relying only on stylistic comparisons of generated images as a proxy. More systematic and quantitative approaches are needed to rigorously assess similarities between models. In future research, we aim to develop more sophisticated metrics and analytical methods to achieve a deeper understanding of model similarity.

## 6 Conclusion

In this paper, we present LTD-Bench, a novel benchmark that enables intuitive and visual evaluation of LLMs by letting them draw. LTD-Bench specifically targets two fundamental aspects: spatial perception and spatial imagination. Our experimental results demonstrate that current LLMs face substantial challenges in both areas, often failing to establish robust bidirectional mappings between linguistic symbols and spatial concepts. Additionally, LTD-Bench provides a promising new avenue for exploring model similarity, offering valuable insights to guide future research in this domain.

## Acknowledgments and Disclosure of Funding

This work was supported by the National Science Fund for Distinguished Young Scholars (No.62025603), the National Natural Science Foundation of China (No. U21B2037, No. U22B2051, No. U23A20383, No. 62176222, No. 62176223, No. 62176226, No. 62072386, No. 62072387, No. 62072389, No. 62002305 and No. 62272401), and the Natural Science Foundation of Fujian Province of China (No. 2021J06003, No.2022J06001).

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

# A    Experiment details

## A.1    Implementation details

For open-source LLMs used in this paper, we utilized the API on the Bailian* platform, which hosts models identical to those on HuggingFace. For GPT-4.1, GPT-4.1-mini and GPT-4o, we use their official API[†]. In our experiments, we set the temperature to 0 for all models. During GPT-4.1-based automated evaluation, it is found that the outputs of GPT-4.1 still exist variance, although the temperature is set as 0. Therefore, we utlize GPT-4.1 to evaluate the outputs of all models by 5 times.

## A.2    More details of human evaluation

For human evaluation, we assigned independent annotators to tasks of different difficulty levels, with 10 annotators dedicated to each level. The annotators cover a diverse range of technical backgrounds, including newly enrolled undergraduates (computer beginners), master's students, and laboratory engineers—ensuring evaluations are not biased toward a single expertise group. For each difficulty level, the final experimental result is determined by averaging the scores from the 10 annotators, which helps mitigate individual subjectivity.

## A.3    Additional evaluation methods experiment

As mentioned in Section 4.1, we employ both human evaluation and GPT-4.1-based automated evaluation for generation tasks at the Easy and Normal levels. Human evaluation provides more accurate and reliable results, as it avoids the hallucinations and occasional misjudgment issues that GPT-4.1 may suffer from. However, human evaluation is labor-intensive and costly, requiring manual inspection of each sample. In contrast, GPT-4.1-based evaluation offers a convenient and fully automated approach, enabling rapid, end-to-end computation of accuracy metrics. For further analysis, we conduct an additional experiment with both evaluation methods. As shown in Figure 2, although GPT-4.1 tends to yield slightly higher accuracy scores due to occasional hallucinations, our experiment confirms that the relative ranking of model performance remains consistent between human and GPT-4.1 evaluations. Therefore, GPT-4.1 can be reliably used for large-scale, automatic evaluation, while human evaluation serves as a more precise but resource-intensive reference.

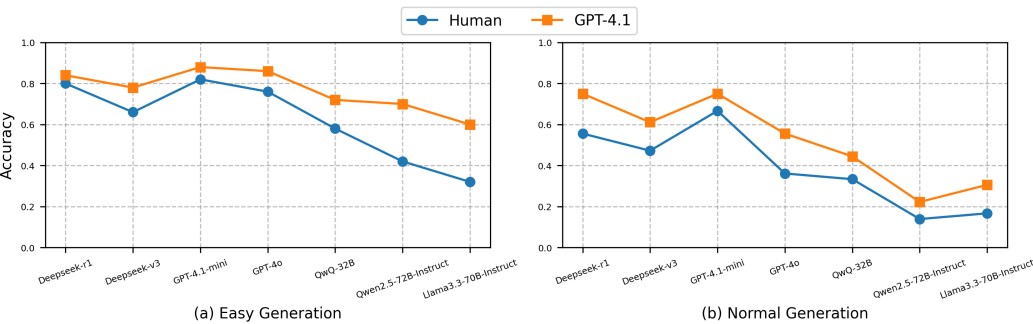

Figure 2: Comparison of human evaluation results and GPT-4.1-based automated evaluation results.

*https://bailian.console.aliyun.com/
†https://platform.openai.com/

## B   Prompt templates

We list all prompt templates used in LTD-Bench here.

> You are a dot matrix drawing robot. I will ask you to draw a specified character on a 0-1 matrix with a specified number of rows and columns.
> Requirements:
> 1. You need to draw the specified character on a 0-1 matrix with a specified number of rows and columns by setting the elements to 1;
> 2. Please strictly follow the following format to output the 0-1 matrix you drew in <Mat></Mat>:
> <Mat>
> mat = []
> </Mat>
>
> Here is the question:
> question

Figure 3: The prompt for the the Easy-level generation task.

> The matrix mat given below is a dot matrix representation of a certain character. Please identify which character it is and fill the answer in the « ».
> Here is the matrix: {matrix}
>
> Here is your answer: « »

Figure 4: The prompt for the Easy-level recognition task.

> You are a code generation robot. You need to generate runnable Python code based on the drawing requirements provided by the user to create the image the user needs.
>
> Requirements:
> 1. Draw the pattern required by the user in a two-dimensional coordinate system, ensuring that the axes are hidden at the end, and do not use the 'Text' or 'TextPath' functions directly for drawing
> 2. The generated image should be saved as "test.jpg"
> 3. Please output in the following format, filling in the generated Python code within the  tags, without adding comments at the beginning or end
> 
> 
>
> Here is the question:
> {question}

Figure 5: The prompt for the Normal-level generation task.

Here is a piece of Python code that draws a certain letter or number. Please determine what the letter or number is and fill the answer in the « ».
Here is the code: {code}

Here is your answer: « »

Figure 6: The prompt for the Normal-level recognition task.

You are a code generation robot. The user will provide drawing requirements for a certain object. You need to generate directly executable Python code according to the drawing requirements to draw the image required by the user.

Requirements:
1. First, analyze the basic features of the drawing object. On this basis, according to the additional drawing requirements proposed by the user, sort out all the feature details that need to be drawn, and conceive how to draw it with Python code
2. Then, generate Python code according to your ideas to draw the image required by the user. Pay attention to the correctness of the library function call
3. Save the drawn image as "test.jpg"
4. Please output in the following format. Fill in all the features and details you need to draw and your ideas in <Thought></Thought>, and fill in the Python code you generated in . Do not add comments at the beginning and end
<Thought>
</Thought>



Here is the question:
{question}

Figure 7: The prompt for the Hard-level generation task.

The following is a dot matrix, where 1 represents fill and 0 represents blank. You need to determine whether the character drawn by the dot matrix is ground truth. If so, output [[Yes]], otherwise output [[No]]. Here is the dot matrix: output matrix

Figure 8: The prompt for evaluating the model outputs on the Easy-level generation task.

Please judge whether the character drawn in the given image is ground truth. If so, output [[Yes]], otherwise output [[No]].

Figure 9: The prompt for evaluating the model outputs on the Normal-level generation task.

{System prompt}
You are an evaluation assistant. Please analyze and score the input image according to the given object and drawing requirements.
Requirements:
1. First determine whether the image can be identified as the given object, then determine whether the image meets the drawing requirements, and finally score based on the analysis
2. The score range (scoring standard) is:

> 0.0: The image cannot identify the object at all
> 0.1: The image can hardly identify the object
> 0.2: The image is difficult to identify the object
> 0.3: The image can barely identify the object, but the main features are blurred and do not meet the drawing requirements
> 0.4: The image can basically identify the object, but does not meet the drawing requirements
> 0.5: The image can identify the object, but only meets a few drawing requirements
> 0.6: The image can identify the object, but a few drawing requirements are not met
> 0.7: The image can identify the object and basically meets all drawing requirements
> 0.8: The image can clearly identify the object and fully meets all drawing requirements, but the painting details and overall aesthetics are poor
> 0.9: The image can clearly identify the object, fully meets all drawing requirements, and the drawing details and overall aesthetics are also excellent
> 1.0: The image can perfectly identify the object, fully meets all drawing requirements, the details are extremely rich, and the overall effect is excellent

3. Strictly follow the format below to output your analysis and final score

> <Analysis>***</Analysis>
> <Score>***</Score>

{User prompt}
Object: {object}
Drawing requirements: {question}

Figure 10: The system prompt and user prompt for evaluating model outputs on the Hard-level generation task.

You are an impartial judge. Please evaluate which two of the three provided images are most similar in style only. Begin your evaluation by comparing the three images and provide a short explanation. Avoid any position biases and ensure that the order in which the images were presented does not influence your decision. After providing your explanation, output your final verdict by strictly following this format: '[[A]]' if the first and second images are more similar, '[[B]]' if if the first and third images are more similar, '[[C]]' if the second and third images are more similar, and '[[D]]' if all three images have different styles.

Figure 11: The prompt for comparing the style similarity of images generated by three different models on the Hard-level generation task.

# C   Case study

In this section, we list several failed cases of different models on generation tasks of three difficulty levels, to further analyze the current limitations in model capabilities.

**Easy level.**   Table 8 shows the failed cases on the generation task at Easy level. We can observe that models often mistakenly generate the characters '>' and 'J' as their mirrored counterparts '<' and 'L' within the dot matrix, revealing their insufficient understanding of basic spatial orientations such as left-right and up-down. Additionally, the way models render the character 'W' in the dot matrix further highlights their limitations in spatial imagination.

**Normal level.**   And for the generation task at Normal level, the failed cases shown in Table 9 more clearly reveal the limitations of the models' spatial capabilities. When questioned with "Draw a blue letter W", QwQ-32B produced an image with the same issue observed in the Easy level earlier: the letter was upside down. Other than that, images generated by other models for other questions listed in the table are even more problematic, featuring completely incorrect outputs with numerous chaotic lines. This suggests that the models may not have a proper understanding of how their actions correspond to spatial states, resulting in outputs that deviate significantly from the intended results. Such shortcomings are critical obstacles for LLMs in achieving a true understanding of the world.

Table 8: The failed cases on the Easy-level generation task.

| Question | Model Outputs | |
|---|---|---|
| *Please draw a character '>'*
*in a 0-1 matrix*
*with 5 rows and 3 columns.* | Deepseek-r1:
 | Deepseek-v3:
 |
| *Please draw a character 'J'*
*in a 0-1 matrix*
*with 5 rows and 3 columns.* | Deepseek-r1:
 | QwQ-32B:
 |
| *Please draw a character 'W'*
*in a 0-1 matrix*
*with 3 rows and 7 columns.* | Deepseek-r1:
 | GPT-4.1-mini:
 |

**Hard level.**   Table 10 further shows the failed cases in Hard level. Firstly, the "Draw a clock with..." case demonstrates that when spatial requirements are introduced, the models tend to perform poorly in easy math and coding tasks. Even advanced models like Deepseek-r1, GPT-4o, and QwQ-32B made mistakes. Specifically, these models may easily understand what "the pointer pointing to 9:30" means, but they often make various errors when asked to translate this understanding into a spatial representation. Further more, in the latter two cases, 'Airplane' and 'Leaf', it is even more apparent that the models' limited spatial imagination, combined with their inadequate ability to map linguistic symbols to spatial entities, leads to these unsatisfactory results.

Through the failed cases and analyses above, we provide a clear and intuitive visualization of the current models' significant shortcomings in spatial capabilities. These findings offer valuable insights for future research on how large language models understand the world.

Table 9: The failed cases on the Normal-level generation task.

| Question | Model Outputs |
|---|---|
| *Draw a purple number 9.* | GPT-4.1-mini:  GPT-4o:  |
| *Draw a blue letter W.* | Deepseek-v3:  QwQ-32B:  |
| *Draw a purple letter J.* | Deepseek-v3:  GPT-4.1-mini:  |

Table 10: The failed cases on the Hard-level generation task.

| Question | Model Outputs |
|---|---|
| *Draw a clock with a round face and the pointer pointing to 9:30* | Deepseek-r1:  GPT-4o:  QwQ-32B:  |
| *Draw a plane, from a bird's-eye view, with two wings and two tails, and the wings should be much longer than the tail* | Deepseek-r1:  GPT-4.1-mini:  QwQ-32B:  |
| *Draw a leaf with veins and irregular jagged edges* | GPT-4.1-mini:  QwQ-32B:  Qwen2.5-72B-Instruct:  |

