# OpenReview forum: "LTD-Bench: Evaluating Large Language Models by Letting Them Draw"
_NeurIPS.cc/2025/Datasets_and_Benchmarks_Track — NeurIPS 2025 Datasets and Benchmarks Track poster_

### Official Review · Reviewer_Z1wF · 2025-06-27

**Rating:** 4
**Confidence:** 1

**Summary:**

This paper proposed a benchmark to test LLMs for visual spatial reasoning. The benchmark consists in a series of prompts asking models (a) to "draw" (i.e., provide code to generate an image of) something requested in the prompt or (b) to describe the output of code generating an image. The idea behind this approach is to have LLM evaluation benchmarks that are easily interpretable even by non experts. Some models seem reasonably good at this, others less so.

**Additional Feedback:**

None.

**Dataset Code Accessibility:**

Yes

**Dataset Code Comments:**

The code snippet at the bottom of this page did not work for me. Code to run the benchmark is available at the authors' link. There is a very minimal readme there, but it would be nice to provide an example of how to run the benchmark on a chosen model. The benchmark data is also available on huggingface. Adding a readme to the huggingface would be useful. Overall, dataset is available but accessibility could be streamlined.

**Ethical Considerations:**

No, there are no or only very minor ethics concerns

**Final Justification:**

The authors have been thorough in replying to my comments, and I therefore increase my score to 4 (Borderline Accept). Overall, this is an interesting approach for evaluating LLMs for "visual" understanding. Remaining limiations: (1) I am still not convinced that the benchmark is constructed in a principled way. (2) I am not fully convinced that there is anything "spatial" in the way the LLMs answer this challenge. It still feels more like a coding challenge than a spatial challenge.

**Limitations Weaknesses:**

Please note that LLM evaluation is really not my area of expertise, so please take these comments with a grain of salt.

Relevance: It is always interesting to look for areas where models struggle, as this provides pointers toward improvments. In this case, though, big models seem to do quite well on the benchmark, so I am not sure that we have learnt.

Methodology:
(a) I am not confident that this approach is principled. It feels more like a collection of semi-random questions asked to LLMs than a principled dataset. If there was some principled methodology for creating the prompts, I did not get it.
(b) LLMs tend to heavily comment code. Was there any care taken to make sure that the code passed to the recognition networks did not contain comments like "plot a green letter Q"? That would make the recognition task trivial.
(c) More generally, I do not think we have strong guarantees that this benchmark really tests spatial reasoning rather than simple code understanding. I am not sure how this could be improved.
(d) While images are indeed interpretable by non-experts, language also is. So I am not sure to what extent this approach leads to more interpretable results. Sure, we can understand an image and see if it makes sense. But we can also do that for text. What makes benchmarks "opaque" is summarize thousands of prompts/responses into a single hard to interpret number. This is what the authors do too, so I do not find this benchmark more interpretable than others. Looking at the benchmark results of Table 3 looks no more and no less interpretable than other benchmarks.

Overstating results: While this benchmark is interesting, I am not sure that it "represents a paradigm shift in the evaluation and understanding of large language models". And there are several other points like this with overstatements.

**Strengths Contributions:**

Novelty: Examining image generating code outputs of LLMs is a relatively novel (I think) way to evaluate models.

Relevance: Creating new ways to evaluate models is always useful.

Presentation: The paper is written adequately (although the language is sometimes quite hyperbolic), and the figures are helpful.

---

> ### Author Rebuttal · Authors · 2025-07-27
>
> Thank you for your constructive and insightful comments. We have carefully addressed each of your concerns as follows:
>
> **1. Relevance**
>
> At first glance, the experimental results in Table 3 might give the impression that many models achieve over 50% accuracy—performance that could seem "quite well". However, it is important to clarify the nature of the tasks: the easy and normal-level tasks are restricted to either drawing characters within constrained grids or composing them via curve combinations in 2D coordinate systems. Based on our understanding of state-of-the-art LLMs, their performance on tasks of this difficulty should be significantly stronger than what we observed. To further support this point, additional visualized examples in Appendix C explicitly demonstrate that even current top-tier LLMs still struggle with such tasks.
>
> **2. Methodology**
>
> **(a)** All prompts used in our experiments were designed by referencing the formats and styles of prompts from classic prior benchmarks (e.g., MT-Bench[1], MM-Vet[2], and MMLU[3]), with targeted adjustments to align with the specific requirements of our tasks.
>
> **(b)** We have conducted a thorough review of the data for the recognition task and removed all comments that might implicitly hint at the correct answers. This ensures that experimental results are not influenced by such confounding factors.
>
> **(c)** Our benchmark evaluates LLMs through intuitive visual assessments, where models must generate or interpret code to complete drawing or recognition tasks. As such, a model’s coding ability is indeed a relevant factor. However, the coding skills required for our benchmark tasks are relatively basic. To clarify the relationship between coding ability and performance on our benchmark, we present evaluation results of several models on mainstream code capability benchmarks in the table below. All data are sourced from model technical reports or official benchmark leaderboards.
>
> | Code Benchmarks | Deepseek-r1 | Deepseek-v3 | GPT-4.1-mini | GPT-4o |
> | :- | :-: | :-: | :-: | :-: |
> | SWE-bench Verified (Resolved) | 49.2 | 42 | 23.6 | 38.8 |
> | Aider-polyglot (Acc.) | 56.9 | 55.1 | 32.4 | 18.2 |
>
> It can be observed that GPT-4.1-mini performs significantly worse than Deepseek-r1 and Deepseek-v3 on both code benchmarks, yet its performance on LTD-Bench (Table 3) is comparable to or even exceeds theirs. Similarly, while Deepseek-v3 lags only slightly behind Deepseek-r1 in coding capability, there is a notable performance gap between them on the normal-level tasks of LTD-Bench. Conversely, Deepseek-v3 marginally outperforms Deepseek-r1 on hard-level tasks. These findings collectively indicate that our benchmark prioritizes the assessment of models' spatial reasoning abilities—provided the models possess basic coding proficiency.
>
> **(d)** Images offer a more intuitive and efficient means of evaluation compared to text. When assessing text-based answers, significant effort and time may be required to compare quality; with images, however, relative performance can often be discerned at a glance. Images also convey more distinct visual impressions, enhancing interpretability— a point further supported by the model similarity analysis in Section 4.3. Specifically, style similarities between generated images allow for straightforward visual comparisons, which in turn reveal similarities between models— a distinction that is far harder to achieve with text.
>
> For evaluation interpretability, we advocate for human assessment, which aligns with our proposed framework of intuitive visual evaluation. This is why Table 2, which presents visualization results from different LLMs, is our key experiment outcome. Additional visualization examples are also provided in Appendix C for reference. Meanwhile, the quantitative results in Table 3 are included just to align with mainstream LLM evaluation practices, ensuring our benchmark also supports end-to-end automatic evaluation.
>
> **3. Overstating results**
>
> We sincerely apologize for any overstated descriptions in the paper. While LTD-Bench innovatively enables visual evaluation of LLMs' spatial capabilities, it does not yet represent "a paradigm shift in the evaluation and understanding of large language models" as previously claimed. We will revise all overstated or inappropriate language in the final version. Moving forward, we aim to enhance LTD-Bench through continuous updates to better fulfill the contributions we initially (and prematurely) highlighted.
>
> **4. Dataset Code**
>
> We apologize for the current lack of detail in the README files of our GitHub and Hugging Face repositories. Due to NeurIPS rebuttal policies, which prohibit authors from updating submitted code repositories during the review period, we will enrich the documentation and improve code readability once the restriction is lifted. This will ensure better usability and support for future users.
>
> ---
>
> [1] Lianmin Zheng, Wei-Lin Chiang, Ying Sheng, Siyuan Zhuang, Zhanghao Wu, Yonghao Zhuang, Zi Lin, Zhuohan Li, Dacheng Li, Eric Xing, et al. Judging llm-as-a-judge with mt-bench and chatbot arena. Advances in Neural Information Processing Systems, 36:46595–46623, 2023.
>
> [2] Weihao Yu, Zhengyuan Yang, Linjie Li, Jianfeng Wang, Kevin Lin, Zicheng Liu, Xinchao Wang, and Lijuan Wang. Mm-vet: Evaluating large multimodal models for integrated capabilities. arXiv preprint arXiv:2308.02490, 2023.
>
> [3] Dan Hendrycks, Collin Burns, Steven Basart, Andy Zou, Mantas Mazeika, Dawn Song, and Jacob Steinhardt. Measuring massive multitask language understanding. arXiv preprint arXiv:2009.03300, 2020.

---

> ### Comment · Reviewer_Z1wF · 2025-08-04
>
> The authors have been thorough in replying to my comments, and I therefore increase my score to 4 (Borderline Accept). Overall, this is an interesting approach for evaluating LLMs for "visual" understanding. Remaining limiations: (1) I am still not convinced that the benchmark is constructed in a principled way. (2) I am not fully convinced that there is anything "spatial" in the way the LLMs answer this challenge. It still feels more like a coding challenge than a spatial challenge.

---

> > ### Author Response · Authors · 2025-08-05
> >
> > Thank you for your valuable suggestions and recognition of our work. Regarding the remaining limitations, we would like to provide further explanations below, hoping to help you better understand our work.
> >
> > **(1)** Concerning the question of whether our benchmark is constructed in a principled way: The overall framework design of LTD-Bench, including data generation, data format, and prompts, is built with reference to the design principles of existing classic benchmarks. We have made some adjustments to the data design to meet the visualization needs of our benchmark, but these adjustments still fall within the scope of classic benchmark design principles. For specific data examples, you can refer to the references provided in our previous rebuttal.
> >
> > **(2)** In our previous rebuttal, we presented the performance of different models on mainstream code capability benchmarks, which does not show a significant positive correlation with their performance on our LTD-Bench. This indicates that our benchmark does not rely solely on the model's code understanding ability. Without excellent spatial reasoning ability, mere code understanding is insufficient for the model to grasp the relative positional relationships between function curves in space. This is precisely why some top-tier LLMs that perform well in terms of coding ability do not achieve satisfactory results on LTD-Bench.

---

### Official Review · Reviewer_1q9A · 2025-06-30

**Rating:** 4
**Confidence:** 5

**Summary:**

LTD-Bench introduces a visually interpretable evaluation framework for LLM spatial reasoning. It focuses on the key issue in the current development of artificial intelligence - the evaluation issue. Particularly, it aims to address the abstract nature of traditional evaluation based on numerical metrics. It requires models to draw (via dot matrices/code) based on textual prompts, exposing gaps in bidirectional language-spatial mapping.

**Additional Feedback:**

1. Why no hard samples for Recognition task?
2. Why does this manuscript choose to evaluate both the ability to generate (draw) and recognize simultaneously? Please explain it in more detail in the main content.
3. Interleaved benchmarks such as ISG-Bench [1], OpenLEAF [2], and OpenING [3] also evaluates both generation and recognition of multimodal large language models, these can also be discussed for enriching the background of the article.

[1] Chen, D., Chen, R., Pu, S., Liu, Z., Wu, Y., Chen, C., ... & Krishna, R. (2024). Interleaved Scene Graph for Interleaved Text-and-Image Generation Assessment. arXiv preprint arXiv:2411.17188.
[2] An, J., Yang, Z., Li, L., Wang, J., Lin, K., Liu, Z., ... & Luo, J. (2024, October). OpenLEAF: A Novel Benchmark for Open-Domain Interleaved Image-Text Generation. In Proceedings of the 32nd ACM International Conference on Multimedia (pp. 11137-11145).
[3] Zhou, P., Peng, X., Song, J., Li, C., Xu, Z., Yang, Y., ... & Zhang, K. (2025). OpenING: A Comprehensive Benchmark for Judging Open-ended Interleaved Image-Text Generation. In Proceedings of the Computer Vision and Pattern Recognition Conference (pp. 56-66).

**Dataset Code Accessibility:**

Yes

**Dataset Code Comments:**

Codes are available at https://anonymous.4open.science/r/LTD-Bench-D324 and dataset is available at  https://huggingface.co/datasets/walktaster/LTD_Bench.

**Ethical Considerations:**

No, there are no or only very minor ethics concerns

**Final Justification:**

The authors addressed most of my main concerns, so I intend to maintain my score, but I would welcome any further discussions to reconsider my rating.

**Limitations Weaknesses:**

1. Please provide more details of human evaluation in the main content. E.g., how many annotators? What is the source and background of recruiting annotators?
2. The benchmark's small dataset (183 data points, Table 1, Line 286-287) and narrow focus on spatial perception/imagination may restrict the generalizability and statistical power of findings.
3. The term "deep reasoning" (Line 250) is used to describe certain models' capabilities without clear definition or supporting evidence within the manuscript.
4. The designed recognition tasks lack ambiguity (e.g., noisy/partial inputs to test robustness).
5. To much typos and grammar errors. For example, there should be a space before the reference symbol at the end of the sentence. This sentence doesn't end with a period: "... patterns into language (perception)" Please double-check the whole manuscript.

**Strengths Contributions:**

1. Innovative Evaluation Paradigm: Shifts evaluation from opaque scores to visual outputs, making limitations (e.g., mirroring errors in Figure 1) accessible to non-experts.
2. Structured Design: Dual-path tasks (generation ↔ recognition) across three difficulty levels (discrete → continuous → real-world objects).
3. Critical Findings: Reveals severe deficiencies in top models (e.g., GPT-4o, Llama3.3) for spatial reasoning, despite strong symbolic performance.

---

> ### Author Rebuttal · Authors · 2025-07-27
>
> Thank you for your constructive and insightful comments. We have carefully addressed each of your concerns as follows:
>
> **1. More details of human evaluation.**
> We apologize for omitting this information in the original submission and will include it in the final version. For human evaluation, we assigned independent annotators to tasks of different difficulty levels, with 10 annotators dedicated to each level. The annotators cover a diverse range of technical backgrounds, including newly enrolled undergraduates (computer beginners), master’s students, and laboratory engineers—ensuring evaluations are not biased toward a single expertise group. For each difficulty level, the final experimental result is determined by averaging the scores from the 10 annotators, which helps mitigate individual subjectivity.
>
> **2. Dataset.**
> We acknowledge that our benchmark currently has limitations in three key areas: the scope of model capabilities evaluated, task diversity, and overall dataset size. We agree that expanding to include additional capability dimensions (e.g., temporal reasoning) or incorporating 3D/dynamic spatial scenarios would enhance the dataset’s generality and statistical validity. However, visual evaluation-oriented datasets pose unique challenges in expansion (e.g., ensuring consistent visual quality and evaluability), making rapid adjustments during the rebuttal phase unfeasible.
>
> During the submission period, we did expand easy- and normal-level data within the existing framework, adding 28 new entries. Unfortunately, this scale remains small, and we identified a risk of introducing new biases with such a limited addition. Additionally, NeurIPS' rebuttle policies also prohibit authors from updating submitted data or code repositories. For these reasons, we have not formally included this data in LTD-Bench for the time being.
>
> Looking ahead, we plan to continuously update LTD-Bench by expanding both task diversity and dataset size, with a focus on addressing the current limitations. This will strengthen the benchmark’s robustness and reliability over time, and we kindly invite you to stay tuned for these developments.
>
> **3. Deep reasoning.**
> Thank you for highlighting this omission. We will add a detailed description of "deep reasoning" to the final version. In our paper, "deep reasoning" refers to the ability to conduct in-depth and thorough logical analysis across multiple interconnected logical nodes [1][2]. A representative model with such capabilities is Deepseek-r1, which builds on Deepseek-v3 and is further trained via extensive reflective reinforcement learning. This training differentiates Deepseek-r1 from Deepseek-v3: the former possesses deep reasoning capabilities, while the latter does not. As for the Deepseek-r1-distill-Llama3.3-70B model, it is a distillation variant of Llama3.3-70B, trained via Supervised Fine-Tuning (SFT) on outputs generated by Deepseek-r1. This process enables it to acquire deep reasoning capabilities comparable to Deepseek-r1. The specific details regarding our utilization of these models are consistent with those described in Appendix A.1.
>
> **4. The designed recognition tasks lack ambiguity.**
> Thank you for this observation. To address this, we have supplemented ablation experiments for the recognition task. Specifically, we introduced controlled disturbances (e.g., deletions or modifications) to the dot matrices or code in the recognition task data—rendering them unable to generate the original correct characters. We then measured how these disturbances affected model performance.
>
> Partial results are shown in the table below, where "0%", "50%", and "100%" indicate the proportion of data subjected to disturbance. As expected, disturbed data becomes nearly unrecognizable, and the experimental results confirm that model accuracy drops sharply as the disturbance ratio increases. This directly validates that our recognition tasks are not trivially solvable and effectively test genuine spatial understanding.
>
> | Disturbance Ratio | Deepseek-r1 | Deepseek-v3 | GPT-4.1-mini | GPT-4o | QwQ-32B | Qwen2.5-72B | Llama3.3-70B |
> | :-: | :-: | :-: | :-: | :-: | :-: | :-: | :-: |
> | 0%(Baseline) | 73.61 | 50.00 | 47.22 | 43.06 | 47.22 | 19.44 | 15.28 |
> | 50% | 38.89 | 26.39 | 25.00 | 22.22 | 23.61 | 12.50 | 9.72 |
> | 100% | 11.11 | 9.72 | 9.72 | 6.94 | 8.33 | 6.94 | 5.56 |
>
> **5. Typos and grammar errors.**
> Thank you for identifying these issues. We apologize for the oversights and will conduct a comprehensive review of the manuscript, correcting all formatting, typographical, and grammatical errors to ensure full compliance with NeurIPS standards.
>
> **6. Why no hard samples for Recognition task.**
> In early stages, we conducted small-scale tests on hard-level recognition tasks. These tasks required models to identify real-world objects from code-generated drawings and describe their detailed features. However, we encountered a critical issue: recognition results were highly biased, with models almost entirely failing to generate correct answers. This stemmed from two limitations: (1) Python code-generated real-world objects are inherently simplistic, and (2) their stylistic representation is highly subjective (e.g., varying interpretations of "a cat face"), making it impossible to define a unique, objective correct answer.
>
> To address this, we recently developed a revised design for hard-level recognition tasks: a multiple-choice format where models select the option most similar to the sample drawing. This reduces ambiguity by constraining responses to predefined candidates. However, implementing this requires two time-intensive steps: writing Python code that reliably renders recognizable real-world objects, and designing valid candidate options. Additionally, NeurIPS' rebuttle policies also prohibit authors from updating submitted data or code repositories. As a result, we are unable to include this design in the current version. We plan to finalize and integrate these hard-level recognition samples in future updates to LTD-Bench, alongside broader task and dataset expansions. We appreciate your understanding and invite you to monitor these developments.
>
> **7. Why does this manuscript choose to evaluate both the ability to generate (draw) and recognize simultaneously.**
> Existing research on LLMs’ spatial reasoning capabilities is scarce, and even multimodal LLMs (MLLMs) exhibit notable deficiencies in this area. Drawing on MLLM spatial reasoning literature, we argue that gaps in spatial reasoning stem primarily from flawed bidirectional mapping between text and space—i.e., converting text descriptions into spatial representations (generation) and interpreting spatial patterns into text (recognition). By decomposing spatial reasoning into these two complementary abilities, we can systematically assess where LLMs struggle: for example, a model might excel at generating drawings from text but fail to recognize similar drawings, indicating a one-sided weakness in spatial reasoning. Evaluating both abilities allows us to pinpoint such gaps, addressing a critical oversight in current research.
>
> **8. Interleaved benchmarks.**
> Thank you for recommending these reference papers. We will thoroughly read them and integrate relevant insights to enrich both the manuscript (e.g., discussions of benchmark design) and LTD-Bench itself (e.g., potential task designs inspired by interleaved evaluation frameworks).
>
> ---
> [1] Chen, Q., Qin, L., Liu, J., Peng, D., Guan, J., Wang, P., ... & Che, W. (2025). Towards reasoning era: A survey of long chain-of-thought for reasoning large language models. arXiv preprint arXiv:2503.09567.
>
> [2] Guo, D., Yang, D., Zhang, H., Song, J., Zhang, R., Xu, R., ... & He, Y. (2025). Deepseek-r1: Incentivizing reasoning capability in llms via reinforcement learning. arXiv preprint arXiv:2501.12948.

---

> > ### Comment · Reviewer_1q9A · 2025-08-05
> > **Thank you for your response**
> >
> > Thank you for your detailed rebuttal and for carefully addressing my previous concerns. Your clarifications on annotator diversity, dataset expansion challenges, and recognition task design are appreciated. To further strengthen the contribution and ensure the robustness of LTD-Bench, I have several additional questions and suggestions:
> >
> > 1. Regarding the proposed multiple-choice format for hard-level recognition, how will you validate that the distractor options are sufficiently challenging and not trivially eliminated by superficial features? Have you considered adversarial design or pilot user studies to calibrate difficulty?
> >
> > 2. Your updated definition for “deep reasoning” is helpful. Could you optionally propose some new special evaluation protocols or framework design that operationalize this concept, beyond referencing prior model architectures?
> >
> > Your rebuttal has addressed my key issues and demonstrates ongoing efforts to improve LTD-Bench. I remain open to reconsidering my score should you be able to provide further clarifications and analyses, especially with respect to the above points.

---

> > ### Author Response · Authors · 2025-08-06
> > **Thank you for your valuable comments**
> >
> > Thank you for your valuable comments. We have carefully considered your questions and suggestions, and our responses are as follows:
> >
> > **(1)** We appreciate your questions and suggestions regarding the data design for the hard-level recognition task. Inspired by the adversarial design you proposed, we can ensure the challenge of this multiple-choice task to a certain extent by designing options with gradually increasing similarity. To elaborate with an example, for a data sample featuring a cat, we can design three levels of difficulty for the multiple-choice options:
> >
> > - For the easy level, options that are completely unrelated in concept (e.g., A. Cat, B. Circle, C. Rectangle, D. Triangle);
> > - For the medium level, options that belong to real-world objects but have relatively large differences in appearance (e.g., A. Cat, B. House, C. Car, D. Apple);
> > - For the high level, options that are of the same type of real-world objects (e.g., A. Cat, B. Dog, C. Mouse, D. Tiger).
> >
> > With this design, we can calculate two indicators: 1. The highest difficulty level each sample that the model can correctly answer; 2. The accuracy rate of the model at each difficulty level. Meanwhile, we will also invite some users to participate in this recognition task to obtain these two indicators for humans, which will serve as a baseline for difficulty reference. This data design method not only allows us to control the difficulty of multiple-choice questions effectively but also enables us to observe the model's spatial reasoning ability under the hard-level recognition task in a more fine-grained manner.
> >
> > **(2)** Inspired by relevant works in the field of long chains of thought, we can design the following new special evaluation protocols or frameworks for models with deep reasoning capabilities. Recent studies [1][2][3][4] have pointed out that models may make errors in the reasoning process output during inference, and asking models to re-verify their own reasoning process can, to a certain extent, allow them to self-correct such errors. Therefore, for models with deep reasoning capabilities, we can ask them to verify whether there are reasoning errors in the reasoning process they initially output, and observe whether the models can identify the errors and thus correct the final answer. Through this method, we can obtain the answers of deep reasoning models after multiple rounds of reflection, rather than only the final answer from the first output like other models. This will enable deep reasoning models to give full play to their advantage of being capable of deep reasoning, allowing us to more comprehensively examine their performance on our benchmark and their spatial reasoning ability under the most ideal conditions.
> >
> > ---
> > [1] Shinn, N., Cassano, F., Gopinath, A., Narasimhan, K., & Yao, S. (2023). Reflexion: Language agents with verbal reinforcement learning. Advances in Neural Information Processing Systems, 36, 8634-8652.
> >
> > [2] Asai, A., Wu, Z., Wang, Y., Sil, A., & Hajishirzi, H. (2024). Self-rag: Learning to retrieve, generate, and critique through self-reflection.
> >
> > [3] Lee, Z., Cao, S., Liu, J., Zhang, J., Liu, W., Che, X., ... & Li, J. (2025). Rearag: Knowledge-guided reasoning enhances factuality of large reasoning models with iterative retrieval augmented generation. arXiv preprint arXiv:2503.21729.
> >
> > [4] Zhang, Z., Ge, T., Liang, Z., Yu, W., Yu, D., Jia, M., ... & Jiang, M. (2024). Learn beyond the answer: Training language models with reflection for mathematical reasoning. arXiv preprint arXiv:2406.12050.

---

> > > ### Comment · Reviewer_1q9A · 2025-08-06
> > > **Response of Reviewer 1q9A**
> > >
> > > The rebuttal has addressed most of my concerns. I encourage the authors to confirm these details in the final version and consider further improvements in future work.

---

### Official Review · Reviewer_uBxk · 2025-07-03

**Rating:** 5
**Confidence:** 4

**Summary:**

LTD-Bench introduces a novel visual evaluation framework for assessing spatial reasoning in Large Language Models (LLMs) by requiring them to generate drawings (via dot matrices or executable code) based on textual descriptions. The benchmark tests bidirectional language-spatial mapping through complementary generation (spatial imagination) and recognition (spatial perception) tasks across three difficulty levels (Easy: discrete grids, Normal: continuous curves, Hard: real-world objects). Experiments reveal significant gaps in current LLMs' spatial reasoning capabilities, even in top-performing models. The visual outputs also enable diagnostic analysis of model similarity.

**Dataset Code Accessibility:**

Yes

**Ethical Considerations:**

No, there are no or only very minor ethics concerns

**Final Justification:**

Before rebuttal, my major concerns are (1) insufficient dataset size; (2) lack of comparisons with a sufficient number of models to check the consistency with human annotations.

In the rebuttal phase, the authors have provided sufficient materials, which include: (1) a comprehensive design to scale up the dataset size; (2) comparison of models like Deepseek-r1, Deepseek-v3, GPT-4.1-mini, GPT-4o, QwQ-32B, Qwen2.5-72B, Llama3.3-70B;

These two additional materials help address my concerns. Considering the fact that dataset scaling is a to-do work, I decided to maintain my score as **5, accept**. I will also change my confidence from **2** to **4**.

**Limitations Weaknesses:**

1. **Dataset Scale**: Relatively small dataset (183 samples), potentially limiting generalizability.
2. **Subjectivity in Hard Tasks**: GPT-4.1-based evaluation of open-ended drawings may inherit model biases and lack human validation.

**Strengths Contributions:**

1. **Novel Evaluation Paradigm**: Transforms abstract metrics into intuitive visual outputs, making spatial reasoning limitations accessible to non-experts.
2. **Comprehensive Task Design**: Systematically evaluates both spatial imagination (text → visual) and perception (visual → text) across progressive complexity levels.
3. **Critical Insights**: Exposes fundamental limitations in LLMs' bidirectional language-spatial mapping, challenging their suitability as "world models."
4. **Diagnostic Utility**: Demonstrates how stylistic similarities in generated images can reveal model architectural relationships (e.g., Qwen family consistency).
5. **Open Resources**: Publicly releases dataset/code (https://anonymous.4open.science/r/LTD-Bench-D324), enhancing reproducibility.

---

> ### Author Rebuttal · Authors · 2025-07-27
>
> Thank you for your constructive and insightful comments. We have carefully addressed each of your concerns as follows:
>
> **1. Dataset Scale**
>
> We acknowledge that our benchmark currently has limitations in three key areas: the scope of model capabilities evaluated, task diversity, and overall dataset size. We agree that expanding to include additional capability dimensions (e.g., temporal reasoning) or incorporating 3D/dynamic spatial scenarios would enhance the dataset’s generality and statistical validity. However, visual evaluation-oriented datasets pose unique challenges in expansion (e.g., ensuring consistent visual quality and evaluability), making rapid adjustments during the rebuttal phase unfeasible.
>
> During the submission period, we did expand easy- and normal-level data within the existing framework, adding 28 new entries. Unfortunately, this scale remains small, and we identified a risk of introducing new biases with such a limited addition. Additionally, NeurIPS' rebuttle policies also prohibit authors from updating submitted data or code repositories. For these reasons, we have not formally included this data in LTD-Bench for the time being.
>
> Looking ahead, we plan to continuously update LTD-Bench by expanding both task diversity and dataset size, with a focus on addressing the current limitations. This will strengthen the benchmark’s robustness and reliability over time, and we invite you to follow these updates.
>
> **2. Subjectivity in Hard Tasks**
>
> Hard-level tasks inherently involve a high degree of subjectivity and openness, which can lead to significant fluctuations in human evaluation results due to factors like individual aesthetic preferences. To address this, we initially adopted GPT-4.1 for standardized automated evaluation. We acknowledge, as you noted, that this approach may introduce its own biases. To mitigate this, we have conducted human evaluations following the same scoring criteria outlined in the prompt used for GPT-4.1. This allowed us to perform a consistency check between human assessments and GPT-4.1-based automated evaluations for hard-level tasks (like Appendix A.2). The results of this check are presented in the table below. As shown, the relative ranking of model performance on hard-level tasks remains consistent between the two evaluation methods—indicating that GPT-4.1 evaluations are reliable for capturing model performance trends, even if absolute scores may vary.
>
> | Evaluation Methods | Deepseek-r1 | Deepseek-v3 | GPT-4.1-mini | GPT-4o | QwQ-32B | Qwen2.5-72B | Llama3.3-70B |
> | :-: | :-: | :-: | :-: | :-: | :-: | :-: | :-: |
> | GPT-4.1 | 63.20 | 66.40 | 71.60 | 48.00 | 42.00 | 40.80 | 35.20 |
> | Human | 57.60 | 60.40 | 66.80 | 41.60 | 37.20 | 35.60 | 28.80 |

---

> > ### Comment · Reviewer_uBxk · 2025-08-05
> > **Thanks for rebuttal**
> >
> > Thanks for the authors' rebuttal.
> >
> > The dataset scale seems to be a remaining issue. I encourage the authors to continue to enrich the dataset at scale.
> >
> > For the table containing different model scores, I am not sure about the meaning of it. Could the authors clarify more about the meaning of each number in the provided table? e.g., what do the scores of Deepseek-r1	Deepseek-v3 mean for instance?

---

> > ### Author Response · Authors · 2025-08-05
> >
> > Thank you for your valuable comment. We are committed to continuously expanding the scale of our dataset and refining LTD-Bench to enhance its robustness and reliability in future iterations.
> >
> > In addition, we sincerely apologize for the lack of detailed explanations regarding the presented table, which may have caused confusion. The scores presented in the table correspond to the performance of each model on hard-level tasks as evaluated by both evaluation methods (all values are percentages, with a maximum score of 100%). For example, Deepseek-r1 achieved a score of 63.20 under GPT-4.1-based automatic evaluation and 57.60 under human evaluation when tackling hard-level tasks.
> >
> > As illustrated in the table, the relative ranking of model performance on hard-level tasks remains consistent between the two evaluation methods. This indicates that GPT-4.1-based automatic evaluation is reliable for capturing model performance trends, even if absolute scores may vary.
> >
> > Notably, this table serves to supplement the missing data on hard-level tasks within the consistency experiment detailed in Appendix A.2 of the manuscript, thereby further validating the consistency between GPT-4.1-based automatic evaluation and human evaluation in assessing hard-level tasks.

---

> > > ### Comment · Reviewer_uBxk · 2025-08-05
> > > **Thanks to authors' response**
> > >
> > > I am also curious about the scores of different models under the easy and the medium level to see if human eval and model eval results (rankings) are still consistent.

---

> > > > ### Author Response · Authors · 2025-08-05
> > > >
> > > > In Appendix A.2 of our paper, the results of the consistency experiments under easy-level and normal-level are provided in detail, which are presented in the form of line charts. This form can show the consistency relationship more intuitively than tables. Therefore, we did not include the results of these experiments in our response. Instead, what is supplemented in our response is only the missing consistency experiment under the hard-level. We sincerely hope that you can understand this approach.​

---

> > > > > ### Comment · Reviewer_uBxk · 2025-08-08
> > > > > **Thanks to authors' response**
> > > > >
> > > > > Thanks to the authors' response. My concerns have been addressed. Considering the fact that the dataset scaling is a to-do work, I decide to maintain my score as **5, accept**, and raise my confidence to **4**.

---

### Official Review · Reviewer_Sn6N · 2025-07-03

**Rating:** 5
**Confidence:** 4

**Summary:**

The paper introduces LTD-Bench, a novel benchmark designed to evaluate the spatial reasoning capabilities of large language models (LLMs) by requiring them to generate visual outputs (e.g., dot matrices or executable drawing code) based on textual instructions. The benchmark assesses both spatial imagination (generation tasks) and spatial perception (recognition tasks) across three difficulty levels (Easy, Normal, Hard). The authors demonstrate that even state-of-the-art LLMs struggle with bidirectional mappings between language and spatial concepts, highlighting a significant gap in current evaluation paradigms. The visual nature of LTD-Bench provides intuitive insights into model limitations, enabling diagnostic analysis and model similarity comparisons.

**Dataset Code Accessibility:**

Yes

**Dataset Code Comments:**

The dataset and code are available at https://anonymous.4open.science/r/LTD-Bench-D324.
Detailed prompts and evaluation protocols are provided in appendices (Appendix B), supporting reproducibility.

**Ethical Considerations:**

No, there are no or only very minor ethics concerns

**Final Justification:**

After reviewing the authors' responses and referencing other reivewers's opinion, I am inclined to raise my score. The authors have acknowledged the limitations of the benchmark and outlined a reasonable plan for expanding the dataset and task diversity in the future. Their approach to evaluating hard-level tasks, which includes human validation alongside GPT-4.1, addresses the subjectivity concerns effectively. Additionally, the exclusion of smaller models, which had shown poor performance due to their limited coding capabilities, was a justified decision to maintain the focus on spatial reasoning. Overall, while some issues remain, the authors have provided sufficient clarification and demonstrated a commitment to addressing the benchmark's limitations, leading me to adjust my score accordingly.

**Limitations Weaknesses:**

Scope: The benchmark focuses narrowly on spatial reasoning, omitting other cognitive dimensions (e.g., temporal reasoning). Expanding tasks to include 3D or dynamic spatial scenarios could enhance generality (Section 5).
    Dataset Size: The current dataset (183 tasks) is relatively small. Increasing task diversity and volume would strengthen statistical validity (Section 5).
    Evaluation Subjectivity: Hard-level tasks rely on GPT-4.1 for scoring, which may introduce bias. Human evaluation for a subset could validate consistency (Section 4.1, Appendix A.2).
    Model Coverage: Experiments are limited to high-performance LLMs. Including smaller models might better delineate capability thresholds (Section 4.1).

**Strengths Contributions:**

Novelty: LTD-Bench addresses a critical blind spot in LLM evaluation by shifting from opaque numerical metrics to visually interpretable outputs, making spatial reasoning limitations transparent (Sections 1, 3).
    Comprehensive Design: The benchmark systematically evaluates spatial cognition through dual-path tasks (generation/recognition) and progressive difficulty levels (Easy/Normal/Hard), offering fine-grained analysis (Section 3.1, Table 1).
    Impact: Reveals significant deficiencies in LLMs' spatial reasoning, challenging assumptions about their capability as "world models" (Section 4.2, Table 3). The visual outputs also facilitate model similarity analysis (Section 4.3, Table 6).
    Reproducibility: The dataset and code are openly available, and the paper provides detailed experimental settings (Appendix A.1) and prompt templates (Appendix B).
    Clarity: Well-organized and clearly written, with informative figures/tables (e.g., Figure 1, Table 2) that illustrate the benchmark’s structure and results.

---

> ### Author Rebuttal · Authors · 2025-07-27
>
> Thank you for your constructive and insightful comments. We have carefully addressed each of your concerns as follows:
>
> **1. Scope and Dataset Size**
>
> We acknowledge that our benchmark currently has limitations in three key areas: the scope of model capabilities evaluated, task diversity, and overall dataset size. We agree that expanding to include additional capability dimensions (e.g., temporal reasoning) or incorporating 3D/dynamic spatial scenarios would enhance the dataset’s generality and statistical validity. However, visual evaluation-oriented datasets pose unique challenges in expansion (e.g., ensuring consistent visual quality and evaluability), making rapid adjustments during the rebuttal phase unfeasible.
>
> During the submission period, we did expand easy- and normal-level data within the existing framework, adding 28 new entries. Unfortunately, this scale remains small, and we identified a risk of introducing new biases with such a limited addition. Additionally, NeurIPS' rebuttle policies also prohibit authors from updating submitted data or code repositories. For these reasons, we have not formally included this data in LTD-Bench for the time being.
>
> Looking ahead, we plan to continuously update LTD-Bench by expanding both task diversity and dataset size, with a focus on addressing the current limitations. This will strengthen the benchmark’s robustness and reliability over time, and we kindly invite you to stay tuned for these developments.
>
> **2. Evaluation Subjectivity**
>
> Hard-level tasks inherently involve a high degree of subjectivity and openness, which can lead to significant fluctuations in human evaluation results due to factors like individual aesthetic preferences. To address this, we initially adopted GPT-4.1 for standardized automated evaluation. We acknowledge, as you noted, that this approach may introduce its own biases. To mitigate this, we have conducted human evaluations following the same scoring criteria outlined in the prompt used for GPT-4.1. This allowed us to perform a consistency check between human assessments and GPT-4.1-based automated evaluations for hard-level tasks (like Appendix A.2). The results of this check are presented in the table below. As shown, the relative ranking of model performance on hard-level tasks remains consistent between the two evaluation methods—indicating that GPT-4.1 evaluations are reliable for capturing model performance trends, even if absolute scores may vary.
>
> | Evaluation Methods | Deepseek-r1 | Deepseek-v3 | GPT-4.1-mini | GPT-4o | QwQ-32B | Qwen2.5-72B | Llama3.3-70B |
> | :-: | :-: | :-: | :-: | :-: | :-: | :-: | :-: |
> | GPT-4.1 | 63.20 | 66.40 | 71.60 | 48.00 | 42.00 | 40.80 | 35.20 |
> | Human | 57.60 | 60.40 | 66.80 | 41.60 | 37.20 | 35.60 | 28.80 |
>
> **3. Model Coverage**
>
> In early-stage experiments, we included several smaller models, and their results (presented in the table below) show extremely poor performance. Analysis of their outputs revealed frequent code errors, which we attribute to the limited coding capabilities of smaller models themselves. Since LTD-Bench is designed primarily to evaluate spatial reasoning abilities—assuming basic coding proficiency—the underperformance of smaller models stems from a fundamental lack of prerequisite skills rather than deficiencies in spatial reasoning. As a result, their test results on LTD-Bench do not provide meaningful insights into the core focus of our benchmark. For this reason, we ultimately decided not to include their experimental results in the paper.
>
> | Models | Easy Generation | Easy Recognition | Normal Generation | Normal Recognition | Hard Generation | Average |
> | :- | :-: | :-: | :-: | :-: | :-: | :-: |
> | Qwen2.5-14B | 24.00 | 8.33 | 5.56 | 11.11 | 18.00 | 13.40 |
> | Qwen2.5-7B | 18.00 | 8.33 | 5.56 | 8.33 | 16.40 | 11.32 |
> | llama3.1-8B | 16.00 | 5.56 | 8.33 | 5.55 | 14.80 | 10.05 |

---

> > ### Comment · Reviewer_Sn6N · 2025-08-05
> >
> > After reviewing the authors' responses and referencing other reivewers's opinion, I am inclined to raise my score. The authors have acknowledged the limitations of the benchmark and outlined a reasonable plan for expanding the dataset and task diversity in the future. Their approach to evaluating hard-level tasks, which includes human validation alongside GPT-4.1, addresses the subjectivity concerns effectively. Additionally, the exclusion of smaller models, which had shown poor performance due to their limited coding capabilities, was a justified decision to maintain the focus on spatial reasoning. Overall, while some issues remain, the authors have provided sufficient clarification and demonstrated a commitment to addressing the benchmark's limitations, leading me to adjust my score accordingly.

---

> > > ### Author Response · Authors · 2025-08-06
> > >
> > > Thank you very much for your recognition and valuable feedback. We truly appreciate your careful review and constructive suggestions, which will greatly inspire our future work. We are committed to further improving the benchmark as planned and will keep striving to enhance its quality.

---

### Comment · Area_Chair_xzXa · 2025-08-03

Hello reviewers,

The author's rebuttal has been posted. Please take some time to read it along with the other reviews. Your feedback on the author's response is highly appreciated to facilitate a productive discussion. Thank you for your time!

---

### Author Response · Authors · 2025-08-09
**Thank all reviewers**

We thank all reviewers for their thoughtful and thorough comments on our paper. In general, we are in complete agreement with the reviewers comments and will make all suggested corrections. We are committed to further improving our benchmark as planned and will keep striving to enhance its quality.

---

### Decision · Program_Chairs · 2025-09-18

**Decision:**

Accept (poster)

**Comment:**

This paper introduces LTD-Bench, a novel and insightful benchmark designed to evaluate the spatial reasoning capabilities of Large Language Models. The core innovation lies in shifting the evaluation paradigm from abstract numerical scores to directly observable visual outputs, requiring models to generate drawings via dot matrices or executable code. The author rebuttal was exceptionally thorough and well-executed, directly leading to a stronger consensus for acceptance. Reviewer Z1wF raised their score, while others (Sn6N, uBxk) strengthened their support in their final justifications. The productive discussion phase demonstrated the authors' commitment and solidified the positive trajectory of the review process. The authors have demonstrated the utility of their benchmark with insightful experiments and have been exemplary in their engagement with the review process, successfully addressing all major concerns with additional experiments and clear future plans. While the dataset size is currently modest, the framework is solid and its initial findings are already significant. LTD-Bench represents a strong and timely contribution to the NeurIPS Datasets and Benchmarks track, and I hope it will inspire valuable future research.